# *When Reviewers Lock Horn*: Finding Disagreement in Scientific Peer Reviews

**Sandeep Kumar†, Tirthankar Ghosal‡, Asif Ekbal†**

†Indian Institute of Technology Patna, India

‡National Center for Computational Sciences, Oak Ridge National Laboratory, USA

†(sandeep_2121cs29,asif)@iitp.ac.in

‡ghosalt@ornl.gov

## Abstract

To this date, the efficacy of the scientific publishing enterprise fundamentally rests on the strength of the peer review process. The journal editor or the conference chair primarily relies on the expert reviewers' assessment, *identify points of agreement and disagreement* and try to reach a consensus to make a fair and informed decision on whether to accept or reject a paper. However, with the escalating number of submissions requiring review, especially in top-tier Artificial Intelligence (AI) conferences, the editor/chair, among many other works, invests a significant, sometimes stressful effort to mitigate reviewer disagreements. Here in this work, we introduce a novel task of automatically identifying contradictions among reviewers on a given article. To this end, we introduce *ContraSciView*, a comprehensive review-pair contradiction dataset on around 8.5k papers (with around 28k review pairs containing nearly 50k review pair comments) from the open review-based ICLR and NeurIPS conferences. We further propose a baseline model that detects contradictory statements from the review pairs. To the best of our knowledge, we make the first attempt to identify disagreements among peer reviewers automatically. We make our dataset and code public for further investigations[1].

## 1 Introduction

Despite being the widely accepted standard for validating scholarly research, the peer-review process has faced substantial criticism. Its perceived lack of transparency (Wicherts, 2016; Parker et al., 2018), sometimes being biased (Stelmakh et al., 2021, 2019), arbitrariness (Brezis and Birukou, 2020), inconsistency (Shah et al., 2018; Langford and Guzdial, 2015), being regarded as a poorly defined task (Rogers and Augenstein, 2020a) and failure to recognize influential papers (Freyne et al., 2010) have

---

[1] https://github.com/sandeep82945/
Contradiction-in-Peer-Review and https://www.iitp.
ac.in/~ai-nlp-ml/resources.html#ContraSciView

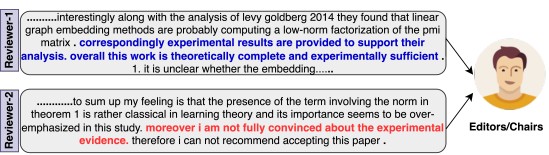

Figure 1: An example of contradiction among reviewers.

all been subjects of concern within the scholarly community. The rapid increase in research article submissions across different venues has caused the peer review system to undergo a huge amount of stress (Kelly et al., 2014; Gropp et al., 2017). The role of an editor in scholarly publishing is crucial (Hames, 2012). Typically, editors or chairs have to manage a multitude of responsibilities. These include finding expert reviewers, assigning review tasks, mediating disagreements among reviewers, ensuring reviews are received on time, stepping in when reviewers are not responsive, making informed decisions, and maintaining communication with authors, among other duties. Simultaneously, they must ensure the validity and quality of the reviews to make an informed decision. However, the complexity of scholarly discourse and inherent subjectivity within research interpretation sometimes leads to conflicting views among peer reviewers (Bornmann and Daniel, 2009; Borcherds and Editor, 2017). For instance, in Figure 1, Reviewer 1 regards the evidence as both strong and sufficient, reinforcing the paper's theoretical soundness. Conversely, Reviewer 2 remains skeptical of this evidence, underscoring their differing perspectives on the soundness of the paper.

Feedback between authors and reviewers can help improve the peer review system (Rogers and Augenstein, 2020b). However, reviewer disagreement can create confusion for authors and editors as they try to address reviewer feedback. There are various guidelines or suggestions that editors consider in order to resolve the conflicting reviews (Borcherds and Editor, 2017). Given this volume, it

is challenging for editors to identify contradictions manually. Our current investigation can assist in pinpointing these discrepancies and help editors make informed decisions.

It is to be noted that this AI-based system aims to aid editors in identifying potential contradictions in reviewer comments. While it provides valuable insights, it is not infallible. Editors should use it as a supplementary tool, understanding that not all contradictions may be captured and manual review remains crucial. They should make decision with careful analysis beyond the system's recommendations.

Our contributions are *three-fold*: 1) We introduce a novel task: identifying contradictions/disagreement within peer reviews. 2) To address the task, we create a novel labeled dataset of around 8.5k papers and 25k reviews. 3) We establish a baseline method as a reference point for further research on this topic.

## 2 Related Work

Artificial Intelligence (AI) has been applied in recent years to the realm of scientific peer review with the goal of enhancing its efficacy and quality (Checco et al., 2021; Ghosal et al., 2022). These applications span a diverse range of tasks, including decision prediction (Kumar et al., 2022; Ghosal et al., 2019), rating prediction (Li et al., 2020; Kang et al., 2018a), sentiment analysis (Kumar et al., 2021; Chakraborty et al., 2020; Kang et al., 2018b), argument mining (Hua et al., 2019; Cheng et al., 2020), and review summarization (Xiong, 2013). In this work, we aim to broaden the scope of AI's usefulness by assisting the editor in determining the disagreement between reviewers. Contradiction detection is not new. Alamri and Stevenson (2015) use linguistic features and support vector machines (SVMs) to detect contradictions in scientific claims, while Badache et al. (2018b) employ review polarity to predict contradiction intensity in reviews. Li et al. (2018) combine sentiment analysis with contradiction detection for Amazon reviews. Meanwhile, Lendvai and Reichel (2016) employ textual similarity features to classify contradictions in rumors, and Li et al. (2017) use contradiction-specific word embeddings for sentence pairs. Tan et al. (2019) propose a dual attention-based gated recurrent unit for conflict detection. Recent advances in Natural Language Inference (NLI) have brought forth a category of textual entailment, categorizing

| Venues | Papers | #Reviews | # Pairs |
|--------|--------|----------|---------|
| ICLR | 5,096 | 14,711 | 14,383 |
| NeurlIPS | 3,486 | 11,114 | 14,114 |
| Total | 8,582 | 25,825 | 28,497 |

Table 1: Dataset statistics

text pairs into entailment, neutral, or contradiction (Chen et al., 2017; Gong et al., 2018). Well-known transformer-based models, such as BERT (Devlin et al., 2019) and its variants leading to state-of-the-art performance.

As far as we know, contradiction detection in peer review has never been studied. Contradiction detection in peer reviews is complex and requires domain knowledge of the subject. It is not straightforward to detect contradictions in peer reviews because reviewers often have different writing styles and approaches to commenting. We believe that our work can significantly contribute to the peer review process.

## 3 Dataset

### 3.1 Dataset Collection

We utilize a subset of 8,582 out of 8,877 papers from the extensive ASAP-Review dataset (Yuan et al., 2021). The dataset comprises reviews from the ICLR (2017-2020) and NeurIPS (2016-2019) conferences. Each review is labeled with eight aspect categories: *(Motivation, Clarity, Soundness, Substance, Originality, Meaningful Comparison, Replicability, and Summary)* along with their sentiment *(Positive/Negative)* except Summary.

### 3.2 Dataset Pre-processing

We define *review* is as a collection of comments/sentences written by one reviewer. Formally, we can represent it as a list:

$$R = \{\text{cmt}_1, \text{cmt}_2, \ldots\}$$

A *review pair* takes two such lists, one from Reviewer1 and one from Reviewer2. It can be represented as:

$$RP = \{R_1, R_2\}$$

Lastly, a *review pair comment* selects one comment from each reviewer and forms a pair. It is a set of such pairs:

$$RPC = \{(\text{cmt from } R_1, \text{cmt from } R_2), \ldots\}$$

To make it easier to annotate, we first create pairs of reviews of papers. Suppose, if there are $n$ number of reviews in a paper then we create $\binom{n}{2}$ pairs resulting in a total of around 28k pairs. Detailed statistics regarding this dataset can be found in Table 1. We follow the contradiction definition of Badache et al. (2018a). According to this definition, a contradiction exists between two review pairs when any review pair comments, denoted as $ra_1$ and $ra_2$, contain a common aspect category but convey opposite sentiments. Therefore, we categorize those review pairs as *no contradiction* if none of their comments share the same aspect with opposing sentiments. This labeling process resulted in 17,095 pairs of reviews. For the remaining review pairs, we compile a list of review pair comments that share the same aspect but express opposing sentiments, and we designate these for human annotation. *Finally, we annotate a total of 50,303 pair of review pair comments.*

## 4 Annotating for Contradiction

### 4.1 Annotation Guidelines

We follow the contradiction definition by De Marneffe et al. (2008) for our annotation process where they define contradiction as: *"ask yourself the following question: If I were shown two contemporaneous documents, one containing each of these passages, would I regard it as very unlikely that both passages could be true at the same time? If so, the two contradict each other"*. We offer multiple examples from different aspect categories that may contain conflicting reviews to assist and direct the annotators. Figure 4 illustrates the flowchart that represents our annotation procedure. For the detailed annotation guidelines, please refer to Appendix A.

### 4.2 Annotator Training

Given the complexity of the reviews and their frequent use of technical terminology, we had six doctoral students, each with four years of experience in scientific research publishing. To facilitate their training, two experts with more than ten years of experience in scientific publishing annotated 1,500 review pairs from a selection of random papers, following our guidelines. Our experts convened to discuss and reconcile any discrepancies in their annotations. The initial dataset comprises 227 pairs with contradictions and 1273 pairs without contradiction comments. We randomly selected 100

review pairs from this more extensive set to train our annotators, ensuring both classes are equally represented. Upon completion of this round of annotation, we reviewed and corrected any misinterpretations with the annotators, further refining their training and enhancing the clarity of the annotation guidelines. To evaluate the effectiveness of the initial training round, we compiled another 80 review pairs from both classes drawn from the remaining review pairs. From the second round on wards, most annotators demonstrated increased proficiency, accurately annotating at least 70% of the contradictory cases.

### 4.3 Annotation Process

We regularly monitored the annotated data, placing emphasis on identifying and rectifying inconsistencies and cases of confusion. We also implemented an iterative feedback system that continuously aimed to refine and improve the annotation process. In cases of conflict or confusion, we consulted experts to make the final decision. Following the annotation phase, we obtained an average inter-annotator agreement score of 0.62 using Cohen's kappa (Cohen, 1960), indicating a substantial consensus among the annotators.

### 4.4 Annotator's Pay

We compensated each annotator based on standard salaries in India, calculated by the hours they worked. The appointment and salaries are governed by the standard practices of our university. We chose not to pay per review pair because the time needed to understand reviews varies due to their complexity, technical terms, and the annotator's familiarity with the topic. Some reviews are also extensive, requiring more time to comprehend. Hence, basing pay on review pairs could have compromised annotation quality. To ensure accuracy and prevent fatigue, we set a daily limit of 6 hours for annotators.

### 4.5 Final Dataset

Figure 2 displays the annotation statistics. We observed that the majority of disagreements among reviewers pertain to the paper's clarity. Such disagreements could stem from differences in reviewers' expertise, domain knowledge (a reviewer unfamiliar with the domain might find it hard to grasp the content), language proficiency (some reviewers might struggle with English, while others are fluent), or interest level (a disinterested reader might

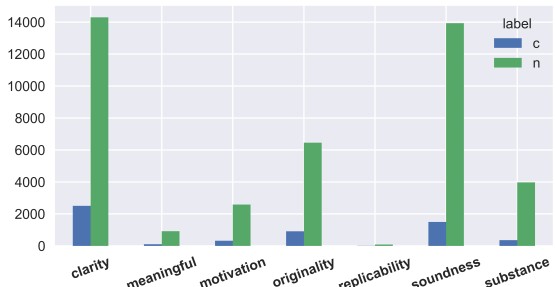

Figure 2: Comparison of Contradiction by Aspect; y axis: count of review comment, x: aspect category, c: contradiction, n: non-contradiction

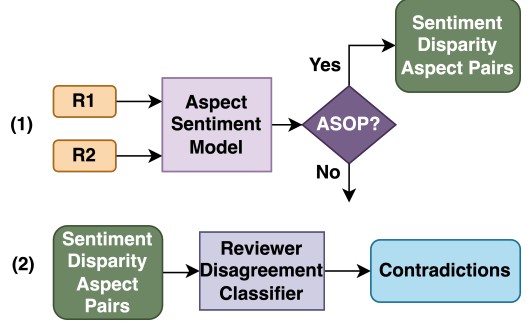

Figure 3: Flowchart of our proposed baseline. Here, R1 and R2 represent the two reviews. The baseline consists of two steps: (1) extracting the SDAPs and (2) determining whether these SDAPs are contradictions ; ASOP stands for "Is Aspect same and sentiment opposite?"

find the paper difficult to engage with). Discrepancies regarding replicability and meaningful comparison are notably fewer, likely because these topics are less frequently commented on.

## 5 Baseline System Description

We describe the flow of our proposed baseline setup through the flowchart in Figure 3. The initial input involves a pair of reviews for a given paper, which are subjected to the Aspect Sentiment Model. This model classifies the aspect and sentiment of each review comment/sentence within the reviews. Subsequently, we identify pairs of comments that, while sharing the same aspects, exhibit differing sentiments; we term it as Sentiment Disparity Aspect Pair (SDAP). As a final step, the SDAPs are then passed to the Contradiction Classifier in order to classify whether these pairs of review comments are contradictory or not. We describe the Aspect Sentiment Model and Contradiction Detection Model in details as follows:

**Aspect Sentiment Model:** Aspect and sentiment in peer review have been studied as a multi-

task model (Kumar et al., 2021). A review pair comment can have different sentiments corresponding to multiple aspect categories. So, we utilize Multi-Instance Multi-Label Learning Network (MIMLLN) (Li et al., 2020) which uniquely identifies sentences as bags and words as instances. It functions as a multi-task model, which performs both Aspect Category Detection (ACD) and Aspect Category Sentiment Analysis (ACSA). Given a sentence and its aspect categories, it predicts sentiment by identifying key instances, and aggregating these to obtain the sentence's sentiment towards each category. We discuss MIMLLN in more detail in Appendix B. We trained MIMLLN on the human-annotated ASAP-Review dataset for our task.

**Reviewer Disagreement Classifier** We use techniques from Natural Language Inference (NLI) sentence pair classification to identify reviewer disagreement, particularly contradictions from Sentence Dependency Pairs (SDPs). Unlike traditional NLI tasks that provide a three-category output of "entailment", "contradiction", and "neutral", we have adjusted the model to a two-category output system: "contradiction" and "non-contradiction". The latter category combines "entailment" and "neutral" labels, as our primary focus is on contradiction detection.

## 6 Results and Analysis

We discuss the implementation details in Appendix C. Table 2 reports the performance of the models when trained on our dataset. The results are of the Reviewer Disagreement Classifier, which is the final step in our proposed approach. We report macro average P, R, and F1 scores as the rarity of the contradiction class is of particular interest for this task (Gowda et al., 2021). RoBERTa large outperforms XLNet large by 0.7 points, RoBERTa base by 4.1 points, XLNet by 4.5 points, and DistilBERT by 7.5 points with respect to F1 score.

In order to analyze how models perform when trained on natural language inference datasets, we trained the models on the ANLI+ALL dataset and evaluated them on our test set. It was found that the models trained by combining datasets, i.e., SNLI, MNLI, FEVER, and ANLI A1+A2+A3, perform the best (Nie et al., 2020). We obtain the highest F1 score of 71.14 F1 with RoBERTa Large. The results show that all the models trained on our dataset outperform those trained on the existing datasets. This large increase in performance can

| Model | P | R | F1 | Acc |
|---|---|---|---|---|
| Finetuned on (ANLI +ALL) | | | | |
| **DistilBERT** (Sanh et al., 2019) | 59.55 | 67.30 | 63.13 | 77.34 |
| **XLNet base**(Yang et al., 2019) | 63.16 | 75.47 | 68.75 | 75.72 |
| **RoBERTa base** (Liu et al., 2019) | 63.15 | 75.63 | 68.90 | 78.53 |
| **XLNet Large** (Yang et al., 2019) | 63.26 | 76.11 | 69.22 | 79.20 |
| **RoBERTa large** (Liu et al., 2019) | **65.32** | **78.03** | **71.14** | **81.34** |
| Trained on our dataset | | | | |
| **DistilBERT** | 69.50 | 70.69 | 70.07 | 87.70 |
| **XLNet base** | 71.13 | 75.68 | 73.05 | 88.07 |
| **RoBERTa base** | 74.49 | 72.43 | 73.40 | 89.83 |
| **XLNet large** | 74.41 | **80.00** | 76.76 | 89.64 |
| **RoBERTa large** | **79.17** | 76.17 | **77.55** | **91.50** |

Table 2: Results of Contradiction Classifier; P: Precision, R: Recall, F1: F1-Score, Acc: Accuracy

be attributed to the differences in reviews and the content of the existing datasets. Scientific peer review comments are written differently from typical human-written English sentences and Wikipedia entries. They contain a more technical style of writing, which is challenging for models to parse when trained on current datasets. Our findings, therefore, highlight the pressing need for innovative datasets in this specific domain. We discuss the results of our proposed intermediate aspect sentiment model in Section D.

Next, we evaluate the performance of RoBERTa Large across the entire process (in evaluation mode). We obtain an accuracy of 88.60% from the Aspect Sentiment Classifier (in determining whether a pair of reviews has any SDAP or not) and a 74.25 F1 score for the Reviewer Disagreement Classifier. We compare our findings with those achieved by the zero-shot Large Language Model, ChatGPT. On the test set, ChatGPT scored an F1 of 64.67 which is 9 points lower than the baseline model, likely due to its lack of explicit training for this specific downstream task. We discuss the prompts and outputs in details in the Appendix E.

We also discuss where our proposed baseline fails (Error Analysis) in the Appendix F. We also utilized the BARD API [2] for our evaluation. We provided identical prompts to both Google BARD and CHATGPT to ensure a fair comparison. We compared our findings with those achieved by Google BARD. BARD scored an F1 of 61.35 on the test set, 12 points lower than the baseline model. This is likely due to its BARD's requirement for more specialized training for this specific task. We

---

[2]https://github.com/dsdanielpark/Bard-API

found that Google BARD registered a higher number of false positives compared to CHATGPT.

## 7 Conclusion and Future Work

In this work, we introduce a novel task to automatically identify contradictions in reviewers' comments. We develop ContraSciView, a review-pair contradiction dataset. We designed a baseline model that combines the MIMLLN framework and NLI model to detect contradictory review pair comments. We found that RoBERTa large performed best for this task with F1 score of 71.14.

Our proposed framework doesn't consider the full review context when predicting contradictions. In the future, we will investigate a system that takes into account the entire review context while detecting contradictory review pair comments. Additionally, we plan to expand our dataset to include other domains and explore the significance of Natural Language Inference for the given task. We also aim to categorize contradictions based on their severity: high, medium, or low.

## Limitations

Our study mainly focuses on identifying "explicit" contradictions. **Explicit Contradictions:** These are clear, direct, and unmistakable contradictions that can be easily identified. For example: The author claims in the introduction that "X has been proven to be beneficial," but in the discussion section, they state that "X has not been shown to have any benefit." One reviewer says, "Figure 2 clearly shows the relationship between A and B," while another reviewer comments, "Figure 2 does not show any clear relationship between A and B."

We do not delve into "implicit" contradictions, which can be hard to detect and can be subjective, making them a topic of debate. **Implicit Contradictions:** These are more subtle and may require deeper understanding or closer examination to identify. It may require the annotators to read the paper and also learn many related works or things to annotate. They are not directly stated but can be inferred from the context or how information is presented. For example: Review 1: "the method lacks algorithmic novelty and the exposition of the method severely inhibits the reader from understanding the proposed idea ." Review 2: "the work presented is novel but there are some notable omissions - there are no specific numbers presented to back up the improvement claims graphs are presented but not specific numeric results - there is limited discussion of the computational cost of the framework presented - there is no comparison to a baseline in which the additional learning cycles used for learning the embedding are used for training the student model ." For instance, Review1 mentions that the method lacks algorithmic novelty, and Review2 acknowledges the work as a novel but points out some notable omissions. This difference in perception is not a direct contradiction, as one reviewer finds the method lacking in novelty, while the other recognizes it as novel but with some omissions.

Additionally, we do not incorporate information from the papers being reviewed, as they cover a wide range of topics and would require many experts from various domains. However, we acknowledge that finding a method to uncover these implicit contradictions in reviews is an intriguing opportunity for future research.

## Ethics Statement

We have utilized the open source peer review dataset for our work. The system sometimes generate incorrect contradictions or overlook certain contradictions. Like other general AI models, this model is not 100% accurate. Editors or Chairs might rely on more than just the contradictions predicted by the model to make decisions. It is important to emphasise that the primary purpose of this system is to assist editors by highlighting potential contradictions between reviewers; this system is only for editors' internal usage, not for authors or reviewers. Especially given their often busy schedules and the myriad of decisions they must make. The contradictions among the reviewers will help editors to identify and initiate a discussion to resolve the confusion between the reviewers and to make an informed decision. Since reviews are frequently extensive and intricate, it is challenging for editors to scrutinise every comment and address conflicts. This system aims to aid them in spotting such contradictions. For instance, if one comment reads, "The paper does not have any new findings," and another reviewer mentions, "The paper is somewhat novel," the editor might not immediately perceive this as contradictory. However, understanding the context and intent behind these comments is vital. If they represent a contradiction, editors can address it using the standard guidelines. The system does not detect all contradictions. If the system fails to identify a contradiction, it does not automatically mean none exists. Given its nature as a general AI system, there is a possibility of it presenting false negatives. Editors relying solely on this system could impact the review process. Any contradictions spotted outside the system's recommendations should also be addressed according to the guidelines. As an AI based model this is prone to errors, the editor/chair are advised to utilize this tool only for assistance and verify the contradiction and analyze carefully before making a decision.

## Acknowledgement

Sandeep Kumar acknowledges the Prime Minister Research Fellowship (PMRF) program of the Govt of India for its support.

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

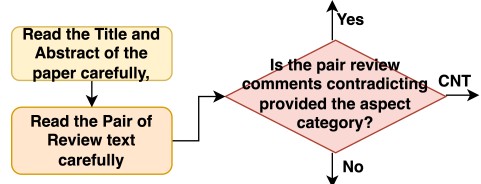

Figure 4: A step-by-step flowchart used by annotators to annotate any pair of review contradiction; Yes:Contradiction, No:Not Contradiction, CNT:Cannot Decide

Xingwei Tan, Yi Cai, and Changxi Zhu. 2019. Recognizing conflict opinions in aspect-level sentiment classification with dual attention networks. In *Proceedings of the 2019 Conference on Empirical Methods in Natural Language Processing and the 9th International Joint Conference on Natural Language Processing, EMNLP-IJCNLP 2019, Hong Kong, China, November 3-7, 2019*, pages 3424–3429. Association for Computational Linguistics.

J.M. Wicherts. 2016. Peer review quality and transparency of the peer-review process in open access and subscription journals. *PLOS ONE*, 11(1).

Wenting Xiong. 2013. Helpfulness-guided review summarization. In *Proceedings of the 2013 NAACL HLT Student Research Workshop*, pages 77–83, Atlanta, Georgia. Association for Computational Linguistics.

Zhilin Yang, Zihang Dai, Yiming Yang, Jaime G. Carbonell, Ruslan Salakhutdinov, and Quoc V. Le. 2019. Xlnet: Generalized autoregressive pretraining for language understanding. In *Advances in Neural Information Processing Systems 32: Annual Conference on Neural Information Processing Systems 2019, NeurIPS 2019, December 8-14, 2019, Vancouver, BC, Canada*, pages 5754–5764.

Weizhe Yuan, Pengfei Liu, and Graham Neubig. 2021. Can we automate scientific reviewing? *CoRR*, abs/2102.00176.

## A  Annotation Guidelines

Recognizing Contradiction in a pair of reviews involves analyzing two pieces of text written by two different reviewers about the same paper. The following steps can be followed:

**Step 1:** Begin by reading the Title and Abstract of the paper to gain an understanding of its subject matter. It is important to read these sections multiple times to grasp the paper's main points, such as its motivation, contributions, and other relevant aspects. If necessary, refer to the paper itself or read related material to enhance your understanding.

**Step 2:** Proceed to read and comprehend the reviews of the paper, focusing on understanding the viewpoints expressed by the reviewers. Take note of their opinions, arguments, and any specific aspects they highlight.

**Step 3:** Based on the reviewer comments, their aspect-sentiment analysis, and the pair of reviews, you should categorize them accordingly. If the reviews contradict each other, indicating opposing viewpoints or conflicting statements, mark it as a contradiction (C). If there is no contradiction or the reviews are unrelated, mark them as non-contradiction (N). In cases where you find it difficult to determine if there is a contradiction or not, mark it as CON (confused).

To decide if the reviews are contradictory, ask yourself the following question: "If I were shown two contemporaneous documents one containing each of these passages, would I regard it as very unlikely that both passages could be true at the same time? If so, the two contradict each other." You should be able to state a clear basis for a contradiction. For example, the following are contradictions:

**R1.** *The motivation of using the method for a very small improvement is not convincing.*

**R2.** *Overall feedback: I found the paper to be well motivated and the proposed approach to be interesting.*

How to make use of the aspect category while annotation?

Given the assigned aspect category, you should utilize the given category to accurately compare these comments. This will help keep your attention strictly on the relevant aspect and refrain from deviating to other topics mentioned in the review text. For example[3]-

**R1:** *However, the results are a visible improvement over JPEG 2000 and I don't know of any other learned encoding which has been shown to achieve this level of performance.*

**R2:** *...the comparison to JPEG2000 is unfortunately not that interesting since that codec does not have widespread usage and likely never will.*

In Review 1, the reviewer values the comparison made with JPEG 2000, considering it a suitable benchmark and a positive aspect of the paper. Conversely, the reviewer in Review 2 argues that the comparison with JPEG 2000 is not compelling due to its limited use and suggests a comparison with WebP would be more relevant. These statements present opposing viewpoints on the relevance of comparing with JPEG 2000 in the paper. While

---

[3]Paper id of the review: ICLR_2017_2

there are many topics discussed in Review 1 and Review 2, it is crucial to concentrate on the remarks concerning the comparison between JPEG and other potential benchmarks when assessing these reviews.

When comparing reviews, keep in mind that one might discuss the subject matter in broad terms, while the other may focus on specific elements.

Consider this example:

**R1:** *The work is original.*

**R2:** *The extension to the partially observable setting is interesting as the proposed form finds a common denominator to multiple estimators, but its underlying idea is not novel.*

In this case, Reviewer 1 comments on the overall originality of the paper, declaring it as novel. On the other hand, Reviewer 2 critiques a specific part of the paper—the extension to the partially observable setting—and declares this particular aspect as not novel. Even though Reviewer 2's comment is more specific, it contradicts the general assertion made by Reviewer 1 about the paper's novelty.

It is important to identify these types of examples as contradictions, even if they may seem to operate at different levels of specificity. The broad comment about the paper's novelty in Review 1 is contradicted by the specific critique in Review 2.

You may find more detailed annotation guidelines in our shared repository.

## B    More details on Aspect Sentiment Model

MIMLLN for Aspect-Category Sentiment Analysis (ACMIMLLN) operates on the assumption that the sentiment of a mentioned aspect category in a sentence aggregates the sentiments of words that indicate that aspect category. In MIMLL, words that indicate an aspect category are termed 'key instances' of that category. Specifically, AC-MIMLLN comprises two components: an attention-based aspect category detection (ACD) classifier and an aspect-category sentiment analysis (ACSA) classifier. Given a sentence, the ACD classifier, as an auxiliary task, assigns weights to the words for each aspect category. These weights signify the likelihood of the words being key instances of their respective aspect categories. The ACSA classifier initially predicts the sentiments of individual words. It then determines the sentence-level sentiment for each aspect category by integrating the respective weights with the word sentiments. The

ACD segment comprises four modules: an embedding layer, an LSTM layer, an attention layer, and an aspect category prediction layer. Similarly, the ACSA segment includes four components: an embedding layer, a multi-layer Bi-LSTM, a word sentiment prediction layer, and an aspect category sentiment prediction layer. In the ACD task, all aspect categories utilize the same embedding and LSTM layers, but they have distinct attention and aspect category prediction layers. For the ACSA task, all aspect categories share the embedding layer, the multi-layer Bi-LSTM, and the word sentiment prediction layer, yet they each have unique aspect category sentiment prediction layers.

## C    Implementation Details

We implemented our system using PyTorch (Paszke et al., 2019), a deep learning framework, and utilized pre-trained transformer models from Hugging Face[4]. The dataset was randomly split into three parts: 80% for training, 10% for validation, and 10% for testing.

For the Aspect Sentiment Model, we conducted experiments with different network configurations during the validation phase. ASAP-Review dataset contains predictions labelled by an aspect tagger on a human-annotated label. We used the 1,000 human-annotated reviews, maintaining the same random split, to train the MIMLLN classifier. Through these experiments, we determined that a batch size of 16 and a dropout rate of 0.5 for every layer yielded optimal performance. The activation function ReLU was used in our model. We trained the model for 15 epochs, employing a learning rate of 1e-3 and cross-entropy as the loss function. To prevent overfitting, we used the Adam optimizer with a weight decay of 1e-3. For the Reviewer Disagreement Classifier, we trained the models using true Sentiment Disparity Aspect pairs with true aspect and sentiment labels. We use a batch size of 16, a maximum length of 280 tokens, and a dropout probability of 0.1. The Adam optimizer was employed with a learning rate of 1e-5. All models were trained on an NVIDIA A100 40GB GPU.

## D    Result of Aspect Sentiment Classifier

We present the results of the Aspect Sentiment Model in Table 3. We conducted experiments using various settings of BERT transformer layers, such as SCIBERT (Beltagy et al., 2019). SCIBERT

---

[4] https://huggingface.co

| Aspect-Category Model \| Scores | | MOT | CLA | SOU | SUB | MEA | ORI | REP |
|---|---|---|---|---|---|---|---|---|
| (Kumar et al., 2021) | $F1_{asp}$ | 0.62 | 0.67 | 0.69 | 0.70 | 0.72 | 0.64 | 0.55 |
| | $F1_{sent}$ | 0.69 | 0.70 | 0.68 | 0.71 | 0.52 | 0.62 | 0.59 |
| MIMLLN (BERT) | $F1_{asp}$ | **0.81** | **0.88** | **0.78** | **0.76** | **0.83** | **0.86** | **0.73** |
| | $F1_{sent}$ | **0.78** | **0.86** | 0.76 | **0.74** | 0.81 | 0.83 | **0.71** |
| MIMLLN(RoBERTa) | $F1_{asp}$ | 0.80 | 0.86 | 0.77 | 0.75 | **0.83** | **0.86** | 0.71 |
| | $F1_{sent}$ | 0.76 | 0.84 | 0.74 | 0.73 | **0.82** | 0.81 | 0.69 |
| MIMLLN(SciBERT) | $F1_{asp}$ | 0.79 | 0.86 | 0.77 | 0.73 | 0.82 | 0.85 | 0.68 |
| | $F1_{sent}$ | 0.77 | 0.85 | 0.75 | 0.72 | 0.81 | 0.81 | 0.67 |
| MIMLLN(SPECTRE) | $F1_{asp}$ | 0.75 | 0.86 | 0.77 | 0.73 | 0.81 | **0.86** | 0.68 |
| | $F1_{sent}$ | 0.74 | 0.85 | **0.76** | 0.73 | 0.80 | **0.84** | 0.66 |

Table 3: Performance of all the models and baselines from the experiments. Here, ($F1_{asp}$ represents the aspect-category F1 scores, $F1_{sent}$ represents the sentiment-category F1 scores)

is a pre-trained language model based on BERT and trained on a large-scale, labeled scientific dataset. Additionally, we experiment with SPECTER (Cohan et al., 2020), which is a model designed for learning representations of scientific papers. This model is based on a Transformer language model pre-trained on citations. We found that the model performed optimally with BERT for aspect classification across all aspect categories.

Furthermore, we compared our results with those of an aspect and sentiment multi-task model (Kumar et al., 2021). This model utilizes a shared BERT transformer layer and employs different feed-forward networks on top as task-specific layers. For each aspect category, pretrained BERT performed the best (RoBERTa performed the same for meaningful comparison).

Regarding aspect-based sentiment, BERT performed the best for the aspects of Motivation, Clarity, Substance, and Replicability. However, for the Soundness category, SPECTER performed better. As for the Meaningful Comparison aspect, RoBERTa showed better performance.

## E   Comparsion with ChatGPT

We conducted a comparative study between our proposed baseline model and ChatGPT and BARD[5], focusing on multi-turn dialogue inputs. We used the OpenAI API[6] for the task. To evaluate the effectiveness of our model, we experimented with various prompts, selecting the most effective one. The prompt chosen is particularly adept at extract-

ing contradictions. We took a single review pair at a time to detect the contradiction. The structure of this prompt is shown in the Figure 5

Here, {review1} and {review2} are placeholders that represent the text from a pair of reviews for which we aim to identify contradictions.

During our testing, we identified many false positive cases in the output of ChatGPT. For example, when presented with the reviews (paper id: ICLR_2018_456):

- **R1:** *The argument of biological plausibility is not justified*

- **R2:** *Moreover the biological plausibility that is used as an argument at several places seems to be false advertising in my view*

ChatGPT incorrectly labeled these reviews as contradictions. However, our baseline system accurately identified that these statements do not present a contradiction.

Additionally, ChatGPT sometimes compares two distinct aspects. For example:

**R1:** *This paper is not well-written.*

**R2:** *The results are reasonable and significant.*

In this instance, the first reviewer is commenting on the clarity of the paper, while the second reviewer is commenting on the substance of the paper. These represent two different perspectives. ChatGPT occasionally fails in such scenarios. However, our proposed baseline incorporates an intermediate aspect sentiment model, which helps reduce these types of errors.

[5]https://bard.google.com/
[6]https://platform.openai.com/docs/api-reference/

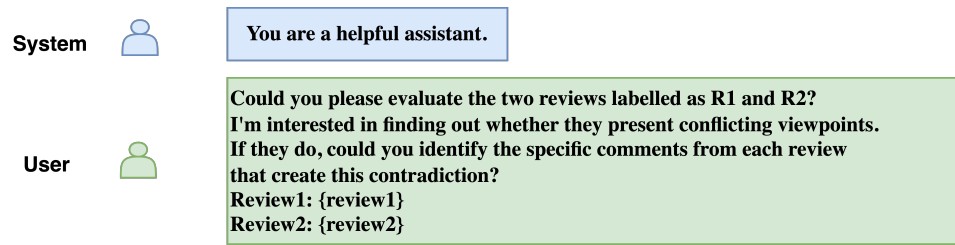

Figure 5: ChatGPT input format

# F Error Analysis

We conducted an analysis of the predictions made by our proposed baseline to identify the areas where it most frequently fails.

## F.1 Lack of Complete Review Context:

Our proposed baseline can sometimes fail in accurately predicting contradictions when the sentences are significantly related to preceding comments within the same review.

Take, for instance, the following comment: *"This is an elegant intuitive algorithm that, to my knowledge, has not appeared in previous literature."*

In this case, the model incorrectly predicts a contradiction with another review comment that discusses a different algorithm negatively. The issue arises from the model's inability to distinguish between the algorithm discussed in this comment and a different algorithm mentioned negatively by another reviewer. While prediction the model needs to learn from previous discussions as well. Such an approach can be considered as a potential area for future work.

## F.2 Issues with Lengthy Comments:

Our proposed baselines sometimes stumble in predicting the right contradictions when dealing with particularly long sentences or complex review comments (significant amount of technical terms or mathematical symbols) which can lead to confusion.

## F.3 Error Propagation

We found error propagation from the first model to the second. To illustrate, consider the following example: Reviewer 1's comment reads: "While it is very interesting to apply adversarial noise in real data, this approach is not clearly motivated or explained." The Aspect sentiment model predicts the aspect for this comment as 'soundness' but misses out on 'motivation'. Reviewer 2's comment states: "In overall, I liked its clear motivation and the simplicity of the method." For this comment, the predicted aspect category by the model is 'motivation'. These two comments provide contrasting viewpoints on the aspect of 'motivation'. Yet, due to the misclassification by the aspect sentiment model, this discrepancy isn not flagged by the second model.