# OpenReview forum: "When Reviewers Lock Horns: Finding Disagreements in Scientific Peer Reviews"
_EMNLP/2023/Conference — EMNLP 2023 Main_

### Official Review · Reviewer_B2WG · 2023-07-28

**Soundness:** 4

**Excitement:**

4: Strong: This paper deepens the understanding of some phenomenon or lowers the barriers to an existing research direction.

**Paper Topic And Main Contributions:**

This paper presents an approach to automatically detect contradictions in reviews for scientific papers. To this end, the authors manually annotated a corpus which in turn was used for the training and evaluation of a baseline system for this novel task.

The main contributions are:
- Introduction of the novel task of automatically identifying contradictions between reviewers' comments on the same paper
- Creation of a novel dataset containing 50k pairs of review comments which were extracted from reviews of about 8.5k papers
- Establishment of a baseline method as a reference point for further research

Main strengths of this paper are the dataset and the evaluation.
1. For the creation of the dataset, the authors collected reviews of the ICLR and NeurIPS conferences. As any other conference, there are already pre-built aspect categories (Motivation, Soundness, Clarity, ...) along with their sentiment. Therefore, a contradiction is located when the reviewers mention the same aspect but with opposing sentiment. These contradiction review comment pairs are then manually annotated.
2. The resulting dataset is used as input to train a baseline system implemented as a pipeline:
(i) Training of an Aspect Sentiment Model to classify the aspects and the sentiment of each review comment. As framework, the authors use the Multi-Instance Multi-Label Learning Network (MIMLLN) of Li et al. (2020) with four models. Based on the outcome of MIMLLN, the authors identify so called Sentiment Disparity Aspect Pairs (SDAP), i.e., pairs mentioning the same aspect but differ in sentiment. Those SDAPs are the input for the next step.
(ii) Training of a Reviewer Disagreement Classifier based on NLI techniques. Unlike in traditional NLI tasks, the output categories are mapped to 'contradiction' and 'non-contradiction' in accordance with the task at hand. As models, the authors used DistilBERT, XLNet base and large as well as RoBERTa base and large. Evaluation shows that RoBERTa large performs best.
3. For completion reason, the authors evaluated the models from (ii) by training them on the ANLI+ALL dataset. Test was performed on the test set from their own dataset. Results show that better scores are obtained when trained on their own dataset. The authors attribute this to the fact that reviews contain a more technical style of writing than e.g. Wikipedia entries.

**Reasons To Accept:**

- Code and dataset is publicly available
- Approach might be interesting for other NLI tasks


**Reasons To Reject:**

None.

**Reproducibility:**

4: Could mostly reproduce the results, but there may be some variation because of sample variance or minor variations in their interpretation of the protocol or method.

**Reviewer Confidence:**

4: Quite sure. I tried to check the important points carefully. It's unlikely, though conceivable, that I missed something that should affect my ratings.

**Typos Grammar Style And Presentation Improvements:**

P2, l112 Missing space 'Gong et al., 2018).Well-known...
P3, description Figure 2: Missing space 'y axis:count', 'c:contradiction', 'n:non-contradiction'
P3, Figure 3: The Figure says 'AOS?' (Purple prism) but description says 'ASOP'
P3, Figure 3: There should be a (1) and (2) in the Figure to match the description
P3, l236: missing space 'Pair(SDAP)'
P4, l241: remove dash (as follows:)
P4, l242-l245: This sentence seems ungrammatical
P4, l245-l247: However, it has not considered that a review can have two different aspects and sentiment.
P4, Table 2: Missing space in second row 'Finetuned on(ANLI+ALL)'
P4, Table 2: Missing space in description 'R:Recall'
P5, l340: 'We have utilizeD the open...'
P9, l777: Missing space '... with the reviews(paper id)...'
P10, l826: Missing space 'comments(significant amount...'

---

> ### Author Rebuttal · Authors · 2023-08-29
>
> Thank you for your careful review and pointing out these formatting and typographical issues in the manuscript. We sincerely appreciate your attention to detail. Here's our response to each of the mentioned concerns:
>
> P2, l112: Thank you for pointing out the missing space. It will be corrected to "Gong et al., 2018). Well-known..."
>
> P3, description Figure 2: We apologize for the oversight. Spaces will be added to make it 'y axis: count', 'c: contradiction', and 'n: non-contradiction'.
>
> P3, Figure 3: The inconsistency between the figure and its description was a typo. We will correct the figure to display 'ASOP' for clarity.
>
> P3, Figure 3: Your suggestion to include (1) and (2) in the Figure to match the description is appreciated. We will make these amendments to enhance clarity.
>
> P3, l236: Thank you for spotting the missing space. It will be corrected to 'Pair (SDAP)'.
>
> P4, l241: The dash will be removed and the text will be updated to "as follows:".
>
> P4, l242-l245: We understand your concerns regarding the sentence's grammar. We will review and rephrase this section to ensure better clarity.
>
> P4, l245-l247: We acknowledge your observation. The sentence will be revised to more clearly convey that a single review can encompass multiple aspects and sentiments.
>
> P4, Table 2: We apologize for the oversight in the spacing. The text will be corrected to 'Finetuned on (ANLI+ALL)'.
>
> P4, Table 2: The missing space in the description will be corrected to 'R: Recall'.
>
> P5, l340: Thank you for pointing out the grammatical error. The sentence will be corrected to "We have utilized the open..."
>
> P9, l777: The missing space will be rectified to "... with the reviews (paper id)...".
>
> P10, l826: We will correct the spacing error to "comments (significant amount...".
>
> In summary, we appreciate your constructive feedback, which will certainly enhance the quality and clarity of our paper. All mentioned issues will be addressed in the revised manuscript.

---

### Official Review · Reviewer_WLX5 · 2023-08-03

**Soundness:** 2

**Excitement:**

3: Ambivalent: It has merits (e.g., it reports state-of-the-art results, the idea is nice), but there are key weaknesses (e.g., it describes incremental work), and it can significantly benefit from another round of revision. However, I won't object to accepting it if my co-reviewers champion it.

**Paper Topic And Main Contributions:**

The topic of this paper is the detection of conflicts/disagreements between different reviewers for the same paper in scientific peer reviews. The main contribution is the formulation of this task and the release of ContraSciView, an annotated dataset of 50,303 pair of review comments. The authors also propose a baseline model to perform this task and compare it to zero-shot ChatGPT.

**Questions For The Authors:**

Q1: The authors use “review pairs” and “review comment pairs” interchangeably. Which is it exactly? How are the reviews chuncked into review comments? Is each review comment processed by the reviewer disagreement classifier individually regardless of the context?

Q2: How are the aspects and sentiments classified in the dataset preprocessing step (line 145 to line 157)? Do the aspects come from the ASAP-Review dataset? Are the authors aware that they are not ground truth labeled by humans but rather predictions made by an aspect tagger?

Q3: Similarly, what data is the aspect sentiment model trained on? Why is the the results from this step not presented and analyzed? Do you consider error propagation from the first model to the second?

Q4: What are the respective definitions of implicit/explicit contradictions. The authors mention that they “do not delve into implicit contradictions” in the limitations section. However, these concepts are not mentioned in the annotation guidelines. This might have been a source of inter-annotator disagreement.

**Reasons To Accept:**

The application of NLP to the peer reviewing process remains an understudied topic that deserves more attention. There is a need to encourage the creation of resources focused on  further exploring this area.

**Reasons To Reject:**

1. The overall motivation and idea is well described. However, there are crucial details missing in the methodology description, as indicated in the questions raised for the authors. These omissions raise concerns about the paper's soundness and validity.

2. The dataset is crawled from OpenReview without obtaining explicit consent from the paper authors and reviewers. This raises potential ethical and legal issues that could have long-term implications.


**Reproducibility:**

3: Could reproduce the results with some difficulty. The settings of parameters are underspecified or subjectively determined; the training/evaluation data are not widely available.

**Reviewer Confidence:**

4: Quite sure. I tried to check the important points carefully. It's unlikely, though conceivable, that I missed something that should affect my ratings.

---

> ### Author Rebuttal · Authors · 2023-08-29
>
> Q1: The authors use “review pairs” and “review comment pairs” interchangeably. Which is it exactly? How are the reviews chuncked into review comments? Is each review comment processed by the reviewer disagreement classifier individually regardless of the context?
> Response: Suppose the review contains two reviews, say R1 and R2. R1 and R2 can have 100,50 sentences each. We call each such pair of sentences for R1 and R2 as review comment pairs. The review pair is full text, and the review pair comment is at sentence level.
>
> Yes, our proposed classifier doesn’t take the full review context while classifier. However, it beats the zero-shot large language models like CHATGPT. We encourage a system that considers full context while classifying future work, which will also be mentioned in the Future Work section. We have also mentioned this in Appendix error analysis A4.1.
>
> Q2: How are the aspects and sentiments classified in the dataset preprocessing step (line 145 to line 157)? Do the aspects come from the ASAP-Review dataset? Are the authors aware that they are not ground truth labeled by humans but rather predictions made by an aspect tagger?
> Response: Yes, we understand that the ASAP-Review dataset contains predictions labelled by an aspect tagger on a human-annotated label. We used those human-annotated labels to train our classifier. We will clarify this in the paper.
> Q3: Similarly, what data is the aspect sentiment model trained on? Why is the the results from this step not presented and analyzed? Do you consider error propagation from the first model to the second?
> Response: Trained on ASAP-Review dataset. Due to the page limit of the paper, we have added the result in the Appendix of the paper, A.2. Yes, there will be error propagation from the first model to the second.  To illustrate, consider the following examples:
> Reviewer 1’s comment reads: “While it is very interesting to apply adversarial noise in real data, this approach is not clearly motivated or explained.” The Aspect sentiment model predicts the aspect for this comment as 'soundness' but misses out on 'motivation'.
> Reviewer 2’s comment states: “In overall, I liked its clear motivation and the simplicity of the method.” For this comment, the predicted aspect category by the model is 'motivation'.
> These two comments provide contrasting viewpoints on the aspect of 'motivation'. Yet, due to the misclassification by the aspect sentiment model, this discrepancy isn't flagged by the second model.
> We thank the reviewer for pointing this out; we will add this analysis in more detail in the Error analysis section of the Appendix.
>
> Q4: What are the respective definitions of implicit/explicit contradictions. The authors mention that they “do not delve into implicit contradictions” in the limitations section. However, these concepts are not mentioned in the annotation guidelines. This might have been a source of inter-annotator disagreement.
>
> Response: Explicit Contradictions: These are clear, direct, and unmistakable contradictions that can be easily identified. For example:
> The author claims in the introduction that "X has been proven to be beneficial," but in the discussion section, they state that "X has not been shown to have any benefit."
> One reviewer says, "Figure 2 clearly shows the relationship between A and B," while another reviewer comments, "Figure 2 does not show any clear relationship between A and B."
>
> Implicit Contradictions: These are more subtle and may require deeper understanding or closer examination to identify. It may require the annotators to read the paper and also learn many related works or things to annotate. They are not directly stated but can be inferred from the context or the way information is presented. For example:
>
> Review 1: “the method lacks algorithmic novelty and the exposition of the method severely inhibits the reader from understanding the proposed idea .”
> Review 2: “the work presented is novel but there are some notable omissions - there are no specific numbers presented to back up the improvement claims graphs are presented but not specific numeric results - there is limited discussion of the computational cost of the framework presented - there is no comparison to a baseline in which the additional learning cycles used for learning the embedding are used for training the student model . "
>
>  For instance, Review1 mentions that the method lacks algorithmic novelty, and Review2  acknowledges the work as a novel but points out some notable omissions. This difference in perception is not a direct contradiction, as one reviewer finds the method lacking in novelty, while the other recognizes it as novel but with some omissions.
>
> Also, if the reviewer mentions some reference to the paper, such as looking at section 3 of the paper, it needs to be clarified, while another reviewer says I find the explanation of the use of contrastive learning we explained. However, section 3 talks about contrastive learning, so both are referring to the same particular. However, it requires reading the paper to determine the contradiction.
> We mentioned in the Annotator Guidelines that “you may find more detailed annotation guidelines on our shared repository”. However, we will also add this point in the Appendix.

---

### Official Review · Reviewer_H8L1 · 2023-08-09

**Soundness:** 4

**Ethical Concerns:**

Yes

**Excitement:**

4: Strong: This paper deepens the understanding of some phenomenon or lowers the barriers to an existing research direction.

**Justification For Ethical Concerns:**

The positive aspect of automating the peer-review process and encouraging reviewer consensus could potentially turn into a negative if editors and meta-reviewers overly depend on the system. This may lead them to accept or reject papers without thoroughly reviewing the comments, raising ethical concerns and violating the ACM Code of Ethics and Professional Conduct, the principle of avoiding harm to the community https://www.acm.org/code-of-ethics.

**Missing References:**

Please check all references:
Anne Borcherds and Managing Editor. 2017. How to deal with conflicting reviewer comments.

**Paper Topic And Main Contributions:**

In this paper, the authors introduce a model designed to detect contradictions in reviewers' feedback. This model is helpful for editors, enabling them to pinpoint contradictory reviews swiftly, thus streamlining the review assessment process and saving time. The proposed model outperforms existing text classification models in its performance. To train the model, the authors curated a dataset comprising pairs of reviews, annotating conflicts to facilitate the training of machine learning algorithms.

**Questions For The Authors:**

Question A: Could you clarify the distinctions between a review, a review pair, and a review pair comment?

Question B: I'm interested in understanding the average, minimum, and maximum number of reviews per paper. It would be beneficial to include more statistical details about the dataset. Can you add such statistics?

Question C: In Figure 2, you display seven aspects of paper reviews classified as contradictions or not. Do all these aspects have corresponding comments, or are they solely multiple-choice fields? For instance, if "soundness" involves multiple-choice responses, how did you identify disagreements?

Question D: How frequently did the annotators encounter instances where they were unable to determine the presence of a contradiction?

Question E: You mentioned splitting the dataset into three parts: 80% for training, 10% for validation, and 10% for testing. Could you elaborate on the choice of a 10% testing set? Was cross-validation employed?

Question F: I'm curious about the specifics of how ChatGPT was employed. Was the API used? Did you utilize the complete set of review pairs or a subset?

Question G: Given your reference to earlier systems for detecting contradictions, would it not be insightful to compare your results with those systems rather than solely testing various models?

Question H: Why was Google BARD not tested?

**Reasons To Accept:**

This research addresses the growing number of submitted manuscripts, potentially automating parts of the peer-review process and promoting reviewer agreement.

The generated dataset holds practical value for the community.

The paper's content is easy to understand.

**Reasons To Reject:**

The positive aspect of automating the peer-review process and encouraging reviewer consensus could potentially turn into a negative if editors and meta-reviewers overly depend on the system. This may lead them to accept or reject papers without thoroughly reviewing the comments, raising ethical concerns and violating the ACM Code of Ethics and Professional Conduct, the principle of avoiding harm to the community https://www.acm.org/code-of-ethics.

The dataset used may exhibit labeling issues. Not all differing comments necessarily contradict each other; some may complement each other instead. Furthermore, the severity of the contradiction is not considered, which could be addressed through categories like high, average, and low levels of contradiction.

The issue of detecting disagreement among reviewers is not new and has been explored before, making the contribution less clear.

Cohen’s kappa of 0.62 is not excellent.

**Reproducibility:**

3: Could reproduce the results with some difficulty. The settings of parameters are underspecified or subjectively determined; the training/evaluation data are not widely available.

**Reviewer Confidence:**

4: Quite sure. I tried to check the important points carefully. It's unlikely, though conceivable, that I missed something that should affect my ratings.

**Typos Grammar Style And Presentation Improvements:**

The statement "We generated a list of review comment pairs for the remaining pairs..." lacks clarity about what is meant by "remaining pairs."

“This labeling process resulted in 17,095”. This doesn’t match the numbers in Table 1.

---

> ### Author Rebuttal · Authors · 2023-08-29
>
> Query 1: The positive aspect of automating the peer-review process and encouraging reviewer consensus could potentially turn into a negative if editors and meta-reviewers overly depend on the system. This may lead them to accept or reject papers without thoroughly reviewing the comments, raising ethical concerns and violating the ACM Code of Ethics and Professional Conduct, the principle of avoiding harm to the community https://www.acm.org/code-of-ethics.
>
> Response: We sincerely appreciate the concerns about the potential over-reliance on an automated peer-review process. Automation is intended to aid and streamline the review process but should not replace the diligent, thoughtful examination traditionally expected of editors and meta-reviewers. We recognize the pivotal role of human judgement, especially in nuanced situations where context and detailed understanding are imperative.
>
> The essence of peer review is thoroughness and integrity, and the ACM Code of Ethics and Professional Conduct underscores the importance of avoiding harm to the community. Hence, while our system seeks to encourage reviewer consensus and improve efficiency, it is paramount that it complements, not supplants, human discernment. We endorse the principle that any tool, automated or otherwise, should be used judiciously, ensuring ethical guidelines and the spirit of peer review remain uncompromised. We will emphasize this perspective in our work to ensure the academic community retains the sanctity of the peer-review process.
>
> Thank you for your constructive feedback regarding our paper. We appreciate your concerns and would like to address them in detail. We understand the ethical implications of the potential inaccuracies of the system. We intend to develop and promote responsible use of our findings, ensuring that users are aware of its limitations. To address this concern more comprehensively, we will expand the Ethical Concerns section to discuss potential misuse, the ramifications of inaccuracies, and the measures we've taken or proposed to mitigate these concerns. A more in-depth discussion will provide readers with a broader perspective on the matter. We will add the following in the ethical section:-
>
> Will our system sometimes generate incorrect contradictions or overlook certain contradictions? Yes, it will. Like other general AI models, this model is not 100% accurate. Editors or Chairs might rely on more than just the contradictions predicted by the model to make decisions. It is important to emphasize that the primary purpose of this system is to assist editors by highlighting potential contradictions between reviewers; this system is only for editors' internal usage, not for authors or reviewers. Especially given their often busy schedules and the myriad of decisions they must make. The contradictions among the reviewers will help editors to identify and initiate a discussion to resolve the confusion between the reviewers and to make an informed decision.
>
> Since reviews are frequently extensive and intricate, it's challenging for editors to scrutinize every comment and address conflicts. This system aims to aid them in spotting such contradictions. For instance, if one comment reads, "The paper doesn't have any new findings," and another reviewer mentions, "The paper is somewhat novel," the editor might not immediately perceive this as contradictory. However, understanding the context and intent behind these comments is vital. If they represent a contradiction, editors can address it using the standard guidelines.
>
> Does the system detect all contradictions? No. If the system fails to identify a contradiction, it doesn't automatically mean none exists. Given its nature as a general AI system, there's a possibility of it presenting false negatives. Editors relying solely on this system could impact the review process. Any contradictions spotted outside the system's recommendations should be addressed according to the guidelines.
>
> We aim to add these points and thoroughly analyze the ethical concerns surrounding our research.
> Thank you for pointing to this critical aspect. We're committed to promoting responsible use of our system in alignment with professional ethics.
>
> Query 2: The dataset used may exhibit labelling issues. Not all differing comments necessarily contradict each other; some may complement each other. Furthermore, the severity of the contradiction is not considered, which could be addressed through categories like high, average, and low levels of contradiction
>
> Response: Thank you for your comment. We acknowledge that not all differing comments inherently exhibit contradiction; some might be complementary. We have added this in the limitation section of the paper.
> Explicit Contradictions: These are clear, direct, and unmistakable contradictions that can be easily identified. For example:
> The author claims in the introduction that "X has been proven to be beneficial," but in the discussion section, they state that "X has not been shown to have any benefit."
> One reviewer says, "Figure 2 clearly shows the relationship between A and B," while another reviewer comments, "Figure 2 does not show any clear relationship between A and B."
> Implicit Contradictions: These are more subtle and may require deeper understanding or closer examination to identify. It may require the annotators to read the paper and also learn many related works or things to annotate. They are not directly stated but can be inferred from the context or how information is presented. For example:
> Review 1: "the method lacks algorithmic novelty and the exposition of the method severely inhibits the reader from understanding the proposed idea ."
> Review 2: "the work presented is novel but there are some notable omissions - there are no specific numbers presented to back up the improvement claims graphs are presented but not specific numeric results - there is limited discussion of the computational cost of the framework presented - there is no comparison to a baseline in which the additional learning cycles used for learning the embedding are used for training the student model ."
>  For instance, Review1 mentions that the method lacks algorithmic novelty, and Review2  acknowledges the work as a novel but points out some notable omissions. This difference in perception is not a direct contradiction, as one reviewer finds the method lacking in novelty, while the other recognizes it as novel but with some omissions.
> Your suggestion regarding grading contradictions based on severity – high, average, or low – is insightful. However, classifying them into three finely delineated categories would require more extensive training for the annotators. The annotation process is already time-consuming and complex, so we decided to keep it as simple as possible for the initial stage. Nevertheless, we see merit in this categorization. While this study serves as foundational work, we aim to integrate such granularity in the future. We will include and elaborate on this in the 'Future Work' section of the paper. Thank you for your valuable feedback.
> Query 3: The issue of detecting disagreement among reviewers is not new and has been explored before, making the contribution less clear.
>
> Response: While we recognize that the detection of disagreements among reviewers in broader domains, such as web sources and customer reviews, is not a novel concept, our research delves into a very specific niche: contradictions within peer reviews. To our knowledge, this specific aspect remains largely untouched in academic studies. It's essential to understand that how peer reviews are penned is inherently different from other review types. This unique writing style, combined with the untouched nature of this research area, positions our dataset and study as both pivotal and essential.
>
> Query 4: Cohen's kappa of 0.62 is not excellent.
>
> Response: This particular annotation requires reading the entire review and understanding the scientific information very carefully. The task is inherently more challenging as compared to other simple annotation tasks. Given the inherently challenging nature of this task, even minor discrepancies in interpretation or understanding can lead to disagreements. This is not a reflection on the competency of our annotators; in fact, they are highly qualified. Instead, it underscores the depth and intricacy of the task at hand.
> Thus, the kappa score we've achieved, while not perfect, is understandable. It's reasonably lower because the task is unreasonably more complex than most.
>
> Questions For The Authors:
>
> Question A: Could you clarify the distinctions between a review, a review pair, and a review pair comment?
>
> Response: A 'review' is a collection of comments/sentences written by one reviewer. Think of it like a list: {comment1, comment2, comment3, ...}.
>
> A 'review pair' is like taking two of these lists - one from Reviewer1 and one from Reviewer2. It looks like this: {Reviewer1's comments, Reviewer2's comments}.
>
> Lastly, a 'review pair comment' picks one comment from each reviewer and pairs them together. So, it's a set of pairs: {(comment from Reviewer1, comment from Reviewer2), ...}.
>
> Question B: I'm interested in understanding the average, minimum, and maximum number of reviews per paper. It would be beneficial to include more statistical details about the dataset. Can you add such statistics?
> Response: We appreciate the suggestion of the reviewer. We recognize the importance of providing comprehensive statistical details regarding our dataset. Understanding the average, minimum, and maximum number of reviews per paper can offer valuable insights into the depth and breadth of our data. We will ensure to include these specifics using a pie chart, along with any other relevant statistics, in the updated version of our paper.
>
> Question C: In Figure 2, you display seven aspects of paper reviews classified as contradictions or not. Do all these aspects have corresponding comments, or are they solely multiple-choice fields? For instance, if "soundness" involves multiple-choice responses, how did you identify disagreements?
>
> Response: If "soundness" involves multiple-choice responses, multiple review pair comments will pass through the classifier for prediction.  For example {( soundness first comment from Reviewer1, soundness first comment from Reviewer2), ( soundness first comment from Reviewer1, soundness second comment from Reviewer2) ...}.
>
> Question D: How frequently did the annotators encounter instances where they were unable to determine the presence of a contradiction?
> Response: Around avg 8% of the time, annotators could not find the presence of contradiction. The experts then resolved it.
>
> Question E: You mentioned splitting the dataset into three parts: 80% for training, 10% for validation, and 10% for testing. Could you elaborate on the choice of a 10% testing set? Was cross-validation employed?
> Response:  We randomly split the data into Training, Testing, and Validation sets. The decision to split the dataset into 80% training, 10% validation, and 10% testing is grounded in a traditional approach to dataset division, aiming to maximize the amount of data available for training while still retaining separate portions for model validation and testing.
>
> Question F: I'm curious about the specifics of how ChatGPT was employed. Was the API used? Did you utilize the complete set of review pairs or a subset?
> Response: We used the OpenAI API for the task. We took a single review pair at a time to detect the contradiction. We have provided a more detailed analysis and prompts used in Appendix A.3 and Figure 5 in the paper.
>
> Question G: Given your reference to earlier systems for detecting contradictions, would it not be insightful to compare your results with those systems rather than solely testing various models?
> Response: Thank you for your feedback. While we have referenced earlier systems for detecting contradictions, our primary focus was on the recent advancements in Natural Language Inference, especially those with contradiction labels. These provide a more pertinent benchmark for our research. That said, we have directly compared our results with those from pre-trained models. We aim to provide a comprehensive understanding of how our approach stands compared to the latest in the field, but we acknowledge the value of juxtaposing our results with earlier systems for a broader perspective. We will add those comparisons as well in our results.
>
> Question H: Why was Google BARD not tested?
> Response: Thank you for bringing up Google BARD. We apologize for the oversight. We recognize the importance of comparing our results with notable models like Google BARD.  We used the BARD API (https://github.com/dsdanielpark/Bard-API) for our evaluation. We provided the same prompt to both Google BARD and CHATGPT for a fair comparison. We compared our findings with those achieved by Google BARD. BARD scored an F1 of 61.35 on the test set, approximately 12 points lower than our baseline model. This is likely due to its need for more explicit training for this downstream task. We found that Google BARD registered more false positives than CHATGPT. We will include a detailed analysis of this in the updated version of our paper.
>
> Missing References:
> Query 1: Please check all references: Anne Borcherds and Managing Editor. 2017. How to deal with conflicting reviewer comments.
> Response: Thank you for highlighting the reference by Anne Borcherds and the Managing Editor from 2017. We will ensure to incorporate this into our paper. Additionally, we'll undertake a thorough review to ensure no other crucial references have been overlooked. We appreciate your diligence in pointing this out and aiding in the enhancement of our work.
>
> Typos Grammar Style And Presentation Improvements:
>
> Query 1: The statement "We generated a list of review comment pairs for the remaining pairs..." lacks clarity about what is meant by "remaining pairs."
> "This labeling process resulted in 17,095". This doesn't match the numbers in Table 1.
>
> Response: We apologize for the confusion due to ambiguous distinctions between a review, a review pair, and a review pair comment. Table 1 presents the statistics for review pairs and the total number of reviews. Among the total review pairs listed in Table 1, some might not include review pair comments that share the same aspect but have conflicting sentiments, suggesting no potential contradictions. Such pairs can be immediately deemed non-contradictory. Therefore, manual labelling was required for only the remaining 17,095 review pairs for some of them.

---

### Official Review · Reviewer_z23x · 2023-08-18

**Soundness:** 2

**Excitement:**

2: Mediocre: This paper makes marginal contributions (vs non-contemporaneous work), so I would rather not see it in the conference.

**Paper Topic And Main Contributions:**

This paper mainly focuses on NLP engineering experiments. This paper addresses the pivotal role of peer review in scientific publishing, highlighting its impact on editorial decisions. The authors present ContraSciView, a new dataset containing review pairs with contradictions extracted from ICLR and NeurIPS conference reviews, totaling around 8.5k papers and 28k review pairs. They also introduce a baseline model for automatically detecting contradictory comments within review pairs, shedding light on the complexities and prospects of identifying contradictions in peer reviews.

**Reasons To Accept:**

The paper tackles a longstanding issue in the peer review process, focusing on the identification of contradictions among reviewers' comments. This problem has persisted over time and significantly impacts the efficacy of the review process. The authors not only shed light on this important problem but also enhance the transparency of their work by sharing their dataset and code, facilitating further research and development in the field.

**Reasons To Reject:**

1. The main concern I have for this paper is the problem definition is not clear. Review contradictions naturally exist and should not be regarded as a good or bad thing, so what is the conclusion of a paper that has fewer contradictions? The conclusion is subjective and individual. For example, other ethical problem, such as the Fairness problem, has a soundness motivation supported, since unfairness is definitely not good.

2. And The new task proposed in this paper is not well-stated. Is there any solution for this task?

So I think this paper is not appropriate for EMNLP because a task should be undisputed and clear.

**Reproducibility:**

4: Could mostly reproduce the results, but there may be some variation because of sample variance or minor variations in their interpretation of the protocol or method.

**Reviewer Confidence:**

4: Quite sure. I tried to check the important points carefully. It's unlikely, though conceivable, that I missed something that should affect my ratings.

---

> ### Author Rebuttal · Authors · 2023-08-29
>
> Query: The main concern I have for this paper is the problem definition is not clear. Review contradictions naturally exist and should not be regarded as a good or bad thing, so what is the conclusion of a paper that has fewer contradictions? The conclusion is subjective and individual. For example, other ethical problem, such as the Fairness problem, has a soundness motivation supported, since unfairness is definitely not good. And The new task proposed in this paper is not well-stated. Is there any solution for this task?
>
> Response: Thank you for your constructive feedback on our paper. We appreciate your concerns regarding the clarity of the problem definition and the potential subjectivity of identifying contradictions in reviews. Allow me to address these concerns comprehensively.
> We intend not to label contradictions as inherently 'good' or 'bad.' Instead, we aim to provide editors with a system that draws attention to potential reviewer inconsistencies. The primary motivation behind our system is to aid editors, given their often demanding roles and the multitude of decisions they face. By highlighting contradictions, our system facilitates editors in pinpointing areas of discussion, enabling them to delve deeper into the nuances and make well-informed decisions. It's essential to reiterate that our system is designed solely for internal usage by editors and is not intended for authors or reviewers.
> Consider an example: if one review suggests, "the paper lacks new findings," while another states, "the paper demonstrates novelty," these might be perceived as contradictory. But the essence lies in understanding the context and intent of these comments. Our system brings such discrepancies to the forefront, allowing editors to analyse, discuss and resolve them based on established guidelines.
>
> Regarding the new task proposed, the paper introduces a fresh perspective. While our exposition could be more precise. In conclusion, our contribution aligns well with the ethos of EMNLP. The goal is to add clarity to the review process, ensuring that the final decisions made by editors are as informed and nuanced as possible. We appreciate your feedback and will work towards enhancing the clarity and precision of our paper by adding the above points. We hope you reconsider its fit upon our revisions.

---

### Meta-Review · Area_Chair_8Zy7 · 2023-09-17

**Recommendation:** 4

**Metareview:**

This paper has got mixed reviews and has also resulted in quite a long and detailed rebuttal.
The content of the paper is, no doubt, interesting and forms a solid short paper which can spur a lot of discussions during the conference (which is always encouraging). Despite having some flaws in the experiment setup, this paper should still make it to the conference.

Now coming to the ethical side of things, the paper has also raised some ethical concerns, and the ethics meta-reviewer has listed down a set of points that the paper must address before it can be published. I remind the authors to carefully take all these into account to prepare the camera-ready version.

---

### Decision · Program_Chairs · 2023-10-07

**Decision:**

Accept-Main

**Comment:**

This paper has got mixed reviews and has also resulted in quite a long and detailed rebuttal.
The content of the paper is, no doubt, interesting and forms a solid short paper which can spur a lot of discussions during the conference (which is always encouraging). Despite having some flaws in the experiment setup, this paper should still make it to the conference.

Now coming to the ethical side of things, the paper has also raised some ethical concerns, and the ethics meta-reviewer has listed down a set of points that the paper must address before it can be published. I remind the authors to carefully take all these into account to prepare the camera-ready version.